# BA2C: Bayesian Advantage Actor Critic for Few Sample Learning using Factor Graph Bayesian Neural Networks

## Abstract

On-policy reinforcement learning (RL) algorithms, such as Proximal Policy Optimization (PPO), are widely used by researchers and practitioners across various tasks. However, these algorithms are known for their lack of sample efficiency, making them challenging to apply when obtaining training samples is costly, particularly in the absence of an effective simulation environment. While some research exists on Bayesian approaches in the context of RL, which promise a better trade-off between exploration and exploitation, to the best of our knowledge, no prior work has explored the implementation of policy-gradient actor-critic algorithms using expectation-propagation for approximate message passing in Bayesian neural networks (BNNs). In this paper, we propose BA2C, an actor-critic algorithm based on networks represented as a factor graph. Since these networks are trained through approximate message passing rather than gradients, we employ a pseudo-target implementation of the policy gradient theorem. We evaluate our algorithm against three popular RL implementations and observe that required training samples can be reduced up to 50% to reach desired levels on certain environments during the early stages of training. Furthermore, our findings indicate that the uncertainty-based evaluation using expectation propagation actually helps, and that our algorithm performs better within the expectation-propagation approximation compared to IVON, a state-of-the-art variational inference algorithm.

## 1 Introduction

**Motivation**   In recent years, major developments have made the field of reinforcement learning (RL) popular. Although the foundations of RL date back to the 1980s and 1990s, the use of deep neural networks (NNs) has led to major breakthroughs, for example, when Atari games could be solved from pixel input only (Mnih et al., 2013), or when the games of Go (Silver et al., 2017b) and Chess (Silver et al., 2017a) could be solved without human knowledge. Apart from games, RL is heavily used for progress in autonomous driving (Kiran et al., 2021), and it has taken an important role in robot control (Singh et al., 2022).

Robot control typically requires RL agents to handle continuous action spaces, a challenge that only a subset of popular RL algorithms can address. Traditionally, actor critic algorithms like Advantage Actor Critic (A2C) and its successor Proximal Policy Optimization (PPO) are used to solve these kinds of problems. They are on-policy algorithms and trained using the policy gradient theorem. On-policy algorithms are popular because of their reliability and high performance. Especially PPO is one of the most widespread RL algorithms today.

**Goal and Focus of this Paper**   However, sample efficiency is a major problem in RL. Even relatively simple problems easily require millions of samples, driving training time and computational effort high. If an accurate and fast simulation environment exists, long training times are inconvenient but manageable. However, if obtaining a large number of samples is expensive or even infeasible, for example, if real-world feedback is needed, the problem of sample efficiency rules out the use of PPO and similar on-policy algorithms. While recent developments have made off-policy algorithms, which use a replay buffer to store and re-use old experiences, more stable, resulting in the development of

Soft Actor Critic (SAC) and TD3, the question remains how on-policy algorithms can be made more sample-efficient, particularly in early training stages.

**Contribution**  This paper analyzes whether Bayesian neural networks (BNN) can be used in on-policy RL algorithms to improve sample efficiency with better exploration and faster learning. Our main contributions are:

- We propose a variant of the A2C algorithm that is based on neural networks (NNs) represented as factor graphs, trained with expectation propagation for message passing (instead of normal NNs), which allows for consistent confidence quantification.
- We adapt the training logic obtained from the policy gradient theorem for the approximate message-passing framework that does not use gradients.
- Our algorithm is applicable to a wide range of continuous control problems.
- In the implemented algorithm, we compare the training performance side-by-side to standard NNs, as they are used in Stable Baselines 3 (Raffin et al., 2021), Ray RLlib (Liang et al., 2017), and Dopamine (Castro et al., 2018), three widespread libraries for RL algorithms.
- For our Bayesian Advantage Actor Critic (BA2C) algorithm, we compare the message-passing framework using expectation propagation against IVON, a state-of-the-art variational inference (VI) algorithm, cf. Shen et al. (2024).
- We show that the use of expectation propagation for training BNNs offers great potential to improve the sample efficiency of RL algorithms. Our code is available on GitHub[1]

In Section 2, we discuss related work. In Section 3, we propose our approach: BA2C. In Section 4, we describe our implementation and design choices. In Section 5, we evaluate our approach against other benchmarks. In Section 6, we summarize and discuss the results obtained.

## 2  RELATED WORK

**RL Methods**  RL is thoroughly introduced by Sutton & Barto (2018). RL methods can be categorized along various dimensions. First, algorithms are split into model-based and model-free; we focus on model-free. Second, some methods rely on simple tabular or linear function approximators, while others employ deep neural networks for complex tasks. Third, algorithms often differ in whether they handle discrete or continuous action spaces.

A key distinction is between off-policy and on-policy algorithms. Off-policy methods, such as Q-Learning (QL) (Watkins & Dayan, 1992) and Deep Q-Learning (DQN) (Mnih et al., 2015), learn from data not necessarily generated by the current policy and typically use a replay buffer. QL was originally designed for discrete actions, but Deep Deterministic Policy Gradient (DDPG) (Silver et al., 2014) extends QL ideas to continuous actions, with Twin Delayed DDPG (TD3) (Fujimoto et al., 2018) and Soft Actor-Critic (SAC) (Haarnoja et al., 2018) improving training stability. By contrast, on-policy methods learn strictly from the current policy. They often build on the policy-gradient theorem (Sutton et al., 1999). REINFORCE (Williams, 1992) is the earliest such algorithm, followed by A2C (Mnih et al., 2016). PPO (Schulman et al., 2017) generalizes A2C and is considered one of the most robust current methods (Huang et al., 2022).

**Bayesian RL Methods**  The area of Bayesian RL has been studied with a variety of methodic contributions. As explained in the survey Ghavamzadeh et al. (2015), Bayesian RL offers the perspective to solve two classic problems in RL, the trade-off between exploration and exploitation, and the option to provide prior knowledge to the agents. The survey gives an overview about several approaches on how to incorporate a Bayesian modeling into classical RL algorithms, starting with multi-armed bandit settings, and discussing TD learning, SARSA and model-based approaches. It also discusses the topic of Partially Observable Markov Decision Processes (POMDPs). However, these methods mainly cover tabular cases or linear models. The ability to scale Bayesian RL to more complex settings is seen as the major challenge.

The survey also discusses an actor-critic algorithm using Bayesian learning, see Ghavamzadeh & Engel (2007). In this work, an explicit posterior was derived for simple tabular problems. In a

---

[1]https://github.com/BayesianAdvantageActorCritic/BayesianAdvantageActorCriticSubmission

subsequent paper Ghavamzadeh et al. (2016), the algorithm was extended by a technique called *Bayesian Quadrature*. More details, proofs, and experimental results were presented. Still, the paper heavily focused on methodic work only with limited scalability.

Achieving higher sample efficiency in exploration was also the motivation for Azizzadenesheli & Anandkumar (2019). The authors utilized Bayesian linear regression for Q-learning. To scale the idea of Bayesian Q-learning to deep neural networks, they used normal neural networks except for the last layer, where they chose Bayesian linear regression instead. They reported to successfully outperform normal deep Q-networks on a variety of Atari games. The idea was taken and adapted, for example, in Ishfaq et al. (2024), Rozanov (2024), and Sasso et al. (2023).

Tasdighi et al. (2024) tackle the common problem of a deficient critic misguiding the actor, resulting in stability problems for actor-critic algorithms. They integrated a probably approximately correct (PAC) Bayesian bound into the SAC algorithm. The algorithm was tested on typical Gym benchmarks, and improvements in sample efficiency were reported, compared to the standard SAC algorithm.

**Factor Graphs & Approximate Message Passing**   Our methodological framework is related to factor graphs and approximate message passing techniques using expectation propagation, see, e.g., Yedidia et al. (2005); Minka (2001). These methods — exploiting the sum product algorithm (Aji & McEliece, 2000) together with a Kullback–Leibler-based approximation of the latent variable marginals — allow to efficiently compute belief distributions of latent variables in Bayesian networks and graphical models (Frey, 1998; Wainwright et al., 2008).

Further, we base our work on an approximate message-passing framework for NNs represented as a factor graph (Sommerfeld et al., 2025) that allows to represent the mean and variance of the weights of a standard NN explicitly. Details of this approach are provided in Appendix B.

**Targeted Research Gap**   Overall, the use of Bayesian techniques to effectively guide the early exploration of AC-based RL methods is still understudied. To the best of our knowledge, there is no existing approach that (i) is generally applicable, (ii) requires only sparse data (and no sampling techniques), (iii) provides confidence for guided exploration in a natural way, and (iv) works for continuous actions. Our aim is to close that gap.

## 3   BAYESIAN ADVANTAGE ACTOR CRITIC (BA2C)

In this section, we propose our algorithm, an adaptation of advantage actor critic with factor graph BNNs trained via approximate message passing. Our algorithm focuses on problems with a continuous action space. A focused explanation of classical A2C is available in the Appendix A. The core motivation is to improve the exploration behavior by using the uncertainty of the BNN to drive exploration. This uncertainty defines the exploration behavior of our agent during training. The intuition is: If the algorithm is confident with its prediction, this is the case because enough data is collected for this situation, and only little exploration will be done. If there is not much data evidence, the algorithm will be unsure and explore more.

**Setup & Notation**   We define our BA2C algorithm for a Markov decision process (MDP) with an $n$-dimensional, continuous state space $\mathcal{S} = \mathbb{R}^n$ and an $m$-dimensional, continuous action space $\mathcal{A} = \mathbb{R}^m$. On this MDP, the BA2C agent will execute actions that are sampled from a stochastic policy $\pi$. The probability density of an action $a \in \mathcal{A}$ in a state $s \in \mathcal{S}$ is

$$\pi_s(a) = P(a|s) = \mathcal{N}\left(a; \mu_s, \mathrm{diag}(\sigma_s^2)\right) \tag{1}$$

and follows a Gaussian distribution, where $\mathcal{N}$ is the probability density of a Gaussian distribution. Its parameters $\mu_s$ and $\sigma_s$ are determined by the actor. In the multi-dimensional case, $\mu_s$ and $\sigma_s$ are vectors, although we only model the individual actions in this action vector to be stochastically independent. Consequently, we write $\mathrm{diag}(\sigma_s^2)$ for the covariance matrix of the multivariate Gaussian distribution. Due to this independence, the actions can also be calculated separately using BNNs with one-dimensional outputs. For simpler notation, let $\Pi_S$ be the (Gaussian) random variable describing the action in state $S$ according to our policy. Like in the standard setup for RL, the policy should maximize the total expected return $J$ defined as

$$J = \mathbb{E}\left[\sum_{i=0}^{\infty} r_i \cdot \gamma^i\right] \tag{2}$$

with $\{S_i\}_{i \in \mathbb{N}}$ as the sequence of visited states, modeled as random variables. For a step $i$, the current state $S_i$ and action $\Pi_{S_i}$ lead to a (random) reward $r_i$ and a new state $S_{i+1}$. This notation refers to an episode of infinite length, the most general case. Of course, it also works for finite episodes (where all rewards are zero from a certain point).

**A2C with Standard Neural Networks**    Appendix A gives a recap on how the classical advantage actor critic works. There, the actor consists of the weights $w_\mu$ and $w_\sigma$[2] to calculate two outputs for a state $s \in \mathcal{S}$:

$$\mu_{w_\mu} : \mathcal{S} \to \mathbb{R}^m; \sigma_{w_\sigma} : \mathcal{S} \to (0, \infty]^m \ . \tag{3}$$

These outputs are used as parameters (mean and standard deviation) for the Gaussian policy. Although $\sigma_{w_\sigma}$ is used as the standard deviation of the Gaussian actor, it has no inherent Bayesian semantics. Instead, it is a supporting output that is part of the policy and trained via the policy gradient theorem, as described in equation 15, see Appendix A.

## 3.1 BA2C with FG-BNNs

In contrast, our proposed algorithm uses NNs represented as a factor graph (FG-BNNs) as function approximators for both actor and critic. These FG-BNNs also get the state as input, and they are designed to have the same architecture regarding nodes and layers. However, they offer a huge advantage by modelling *distributions* over the function values of actor and critic (where the distribution results from the inherent uncertainty due to a limited training set rather than the uncertainty of the environment only). The factor-graph framework works inherently with Gaussian distributions as approximations, passing natural parameters of normal distributions from layer to layer.

The fact that the framework works with Gaussian distributions is a natural fit to the popular modelling of Gaussian policies for continuous action spaces. The optimal action in a state itself is modeled as an approximate latent variable $\hat{B}_{\theta_a}(s)$ in the factor graph with parameters $\theta_a$. For a state $s$, the factor graph calculates a belief on $\hat{B}_{\theta_a}(s)$, a Gaussian approximation of its predictive posterior described by a mean vector $\mu_{\theta_a}(s)$ and the standard deviations of the uncertainty of these means $\sigma_{\mu_{\theta_a}}(s)$. Formally, we get a belief distribution for the best action $a \in \mathcal{A}$ with density, $s \in \mathcal{S}$,

$$f_{\hat{B}_{\theta_a}(s)}(a|s) = \mathcal{N}\left(a; \mu_{\theta_a}(s), \mathrm{diag}\left(\sigma_{\mu_{\theta_a}}(s)^2\right)\right) \ . \tag{4}$$

While $\mu_{\theta_a}$ of these Gaussian output distributions can be viewed as the point-estimates, the standard deviations express the uncertainty resulting from a lack of training data. This standard deviation is small if there is a lot of supporting training data in the state-space area, and it should be large if there is not much training data similar to this sample.

From a high-level perspective, the algorithm appears quite similar to standard A2C (both use a Gaussian policy for a continuous action space with parameters obtained from a parameterized network). But the concept of obtaining the standard deviation to drive the exploration is different.

**BA2C Critic Training**    While the critic in case of standard NNs just gives a point estimate, the critic in BA2C is an FG-BNN with parameters $\theta_c$. It gives a Gaussian approximation on the state value, representing the uncertainty based on the lack of training data. Let $V_{\theta_c}$ be this random variable. It depends on the state $s$, where the probability density is, $v \in \mathbb{R}$, $s \in \mathcal{S}$,

$$f_{\hat{V}_{\theta_c}}(v|s) = \mathcal{N}\left(v; \mu_{\theta_v}(s), \sigma_{\mu_{\theta_v}}(s)^2\right) \ . \tag{5}$$

Here, $\mu_{\theta_v}(s)$ is the mean parameter and $\sigma_{\mu_{\theta_v}}(s)$ the standard deviation parameter, that are retrieved from the critic FG-BNN. Note that this variance $\sigma_{\mu_{\theta_v}}(s)^2$ may not be mixed up with the distribution of actual returns that will be obtained from this state. Instead, it is a measure for the certainty of the estimation of the state value (i.e. a mean). BA2C's critic is trained via bootstrapping, just like the standard A2C. Hence, for the training of the critic in state $s$, we take a sample $a$ from the approximate, estimated action distribution $\hat{B}_{\theta_c}$. This action is executed in the environment, a reward

---

[2]In most implementations, an actor network shares the same parameters for the mean and variance prediction, except for the last layer.

$r_{s,a}$ is observed, and the new state $s'$ retrieved. With this information and recursive use of our critic, we calculate a target estimation $\mathcal{R}(s,a)$ as, $a \in \mathcal{A}$, $s \in \mathcal{S}$,

$$\mathcal{R}(s,a) = r_{s,a} + \gamma V_{\theta_c}(s') \,. \tag{6}$$

Note that $\mathcal{R}(s,a)$ is a Gaussian random variable because $V_{\theta_c}(s)$ is a Gaussian; only linear scaling and addition with real numbers is used. The mean can be easily calculated as

$$\mathbb{E}\left[\mathcal{R}(s,a)\right] = \mathbb{E}\left[r_{s,a} + \gamma V_{\theta_c}(s)\right] = r_{s,a} + \gamma \mathbb{E}\left[V_{\theta_c}(s)\right] = r_{s,a} + \gamma \mu_{\theta_v}(s) \,, \tag{7}$$

and the variance is

$$\mathbb{V}\left[\mathcal{R}(s,a)\right] = \mathbb{V}\left[r_{s,a} + \gamma V_{\theta_c}(s)\right] = \gamma^2 \mathbb{V}\left[V_{\theta_c}(s)\right] = \gamma^2 \sigma_{\mu_{\theta_v}}(s)^2 \,. \tag{8}$$

When training the parameters $\theta_c$ of the factor graph, these two momentums are set for the regression factor. The continuous calculation with uncertainties takes elegantly account for the fact that the targets are also uncertain, especially during early training.

**Advantages as Random Variables**    In a state $s \in \mathcal{S}$, let $a \in \mathcal{A}$ be the action that was taken. Now, we define $A(a,s) := \mathcal{R}(s,a) - V_{\theta_c}(s)$ as the advantage.[3] It is defined analogously to normal A2C, but with random variables. The interpretation of the reward is if the execution of action $a$ did improve the situation compared to what is expected in state $s$. It is also Gaussian distributed with mean

$$\begin{aligned}\mathbb{E}\left[A(a,s)\right] &= \mathbb{E}\left[\mathcal{R}(s,a) - V_{\theta_c}(s)\right] = \mathbb{E}\left[\mathcal{R}(s,a)\right] - \mathbb{E}\left[V_{\theta_c}(s)\right] \\ &= r_{s,a} + \gamma \mu_{\theta_v}(s) - \mu_{\theta_v}(s) = r_{s,a} - \mu_{\theta_v}(s)(1-\gamma)\end{aligned} \tag{9}$$

due to equation 7. With equation 8 and since we assume independence the variance amounts to

$$\mathbb{V}\left[A(a,s)\right] = \mathbb{V}\left[\mathcal{R}(s,a) - V_{\theta_c}(s)\right] = \mathbb{V}\left[\mathcal{R}(s,a)\right] + \mathbb{V}\left[V_{\theta_c}(s)\right] = \gamma^2 \sigma_{\mu_{\theta_v}}(s)^2 + \sigma_{\mu_{\theta_v}}(s)^2 \,. \tag{10}$$

## 3.2 BA2C Actor Training via Pseudo Targets

An important change from normal A2C to its Bayesian sibling is necessary for the training of the actor. It cannot make direct use of the policy gradient, since the message-passing framework for our FG-BNN does not train via gradients. To tackle this problem, our algorithm makes use of *pseudo targets*. In a state $s \in \mathcal{S}$, these pseudo targets $T_{a,s}$ are obtained as a shift of the currently believed optimal action $\hat{B}_{\theta_a}(s)$ towards the direction given by the policy gradient theorem, scaled by a learning rate $\alpha$. Let $a$ be the actually taken action which is a sample from the belief distribution, and let $A(a,s)$ be the advantage. $\mathcal{N}\left(a; \mu_{\theta_a}(s), \mathrm{diag}\left(\sigma_{\mu_{\theta_a}}(s)^2\right)\right)$ is the probability density of the taken action under this belief, described by the two moments $\mu_{\theta_a}(s)$ and $\sigma_{\mu_{\theta_a}}(s)$. Note that we are only shifting the mean parameter in BA2C. Also, we are taking the derivative just with respect to the mean $\mu_{\theta_a}$, not the parameters $\theta_a$. Then, the pseudo targets $T_{a,s}$ are defined as, $s \in \mathcal{S}$, $a \in \mathcal{A}$,

$$\begin{aligned}T_{a,s} &= \hat{B}_{\theta_a}(s) + \alpha \cdot A(a,s) \cdot \quad \nabla_{\mu_{\theta_a}} \log\left[\mathcal{N}\left(a; \mu_{\theta_a}(s), \mathrm{diag}\left(\sigma_{\mu_{\theta_a}}(s)^2\right)\right)\right] \\ &= \hat{B}_{\theta_a}(s) + \alpha \cdot A(a,s) \cdot \quad \nabla_{\mu_{\theta_a}} \log\left[\frac{1}{\sqrt{2\pi}\sigma_{\mu_{\theta_a}}(s)} \cdot \exp\left[-\frac{(a - \mu_{\theta_a}(s))^2}{2\sigma_{\mu_{\theta_a}}(s)^2}\right]\right] \\ &= \hat{B}_{\theta_a}(s) + \alpha \cdot A(a,s) \cdot (a - \mu_{\theta_a}(s))/(2\sigma_{\mu_{\theta_a}}(s)^2) \,.\end{aligned} \tag{11}$$

Fortunately, the derivative of the log probability density of a normal distribution with respect to the mean parameter is quite simple. Because $\hat{B}_{\theta_a}(s)$ and $A(a,s)$ are Gaussian random variables, the other variables are constants, and Gaussian distributions are closed under linear scaling and addition, $T_{a,s}$ is also a Gaussian random variable. We can calculate the mean parameter vector $\mathbb{E}\left[T_{a,s}\right]$ in a straightforward way with the results from equation 11 and equation 9:

$$\begin{aligned}\mathbb{E}\left[T_{a,s}\right] &= \mathbb{E}\left[\hat{B}_{\theta_a}(s) + \alpha \cdot A(a,s) \cdot \frac{a - \mu_{\theta_a}(s)}{2\sigma_{\mu_{\theta_a}}(s)^2}\right] = \mathbb{E}\left[\hat{B}_{\theta_a}(s)\right] + \alpha \cdot \mathbb{E}\left[A(a,s)\right] \cdot \frac{a - \mu_{\theta_a}(s)}{2\sigma_{\mu_{\theta_a}}(s)^2} \\ &= \mu_{\theta_a}(s) + \alpha \cdot (r_{s,a} - \mu_{\theta_v}(s)(1-\gamma)) \cdot (a - \mu_{\theta_a}(s))/(2\sigma_{\mu_{\theta_a}}(s)^2) \,.\end{aligned} \tag{12}$$

---

[3] $V_{\theta_c}(s)$ and $\mathcal{R}(s,a)$ as defined in the previous section

With the reward $r_{s,a}$, this can be explicitly calculated in the training loop. Further, with zero covariance due to the assumed independence between the belief distribution and the advantage distribution, we get a very similar calculation for the variance:

$$\mathbb{V}\left[T_{a,s}\right] = \mathbb{V}\left[\hat{B}_{\theta_a}(s) + \alpha \cdot A(a,s) \cdot \frac{a - \mu_{\theta_a}(s)}{2\sigma_{\mu_{\theta_a}}(s)^2}\right] = \mathbb{V}\left[\hat{B}_{\theta_a}(s)\right] + \alpha^2 \cdot \mathbb{V}\left[A(a,s)\right] \cdot \left(\frac{a - \mu_{\theta_a}(s)}{2\sigma_{\mu_{\theta_a}}(s)^2}\right)^2$$

$$= \sigma_{\mu_{\theta_a}}(s)^2 + \alpha^2 \cdot \left(\gamma^2 \sigma_{\mu_{\theta_v}}(s)^2 + \sigma_{\mu_{\theta_v}}(s)^2\right) \cdot \left((a - \mu_{\theta_a}(s))/(2\sigma_{\mu_{\theta_a}}(s)^2)\right)^2,$$

(13)

using equation 11 and equation 10. These two parameters define the training target for the training of the actor.

## 4 IMPLEMENTATION

In this section, we give an overview of the most important aspects of the implementation. Detailed explanations of the design choices are provided in Appendix C.1. The BA2C algorithm is implemented in Julia. In our BA2C implementation, the FG-BNNs are accessed via a light API. This enables us to exchange FG-BNNs for other types of BNNs or standard NNs while keeping the rest of the BA2C algorithm, including the pseudo-target calculation, unchanged. Further, this allows us to distinguish between the effects of the FG-BNNs and those of the design decisions for BA2C. We are making use of this in Sections 5.1, 5.2, and 5.3. In Appendix C.2, we explain a few design choices for greater stability, such as independence for one training batch, clipping of the pseudo targets, and decreasing the learning rate. Review Appendix C.3 for notable differences in comparison to standard NNs and Appendix C.4 for an explanation of new hyperparameters that come with the selection of FG-BNNs. There, a comprehensive overview of the selected hyperparameters is given in Table 1.

## 5 EXPERIMENTS & EVALUATION

We use different experiments to evaluate the strengths and weaknesses of our Bayesian approach. Within our implementation of the actor critic agent, we are going to evaluate the performance against normal NNs and IVON to justify the choice of the message-passing framework. Afterwards, we will compare the implementation against three of the most widespread implementations of PPO.

**Evaluation Methods**   The focus of our evaluation is to assess if the Bayesian approach has an advantage in the early phasis of training. To discuss that, we are using learning curves, a standard way of evaluating RL algorithms. For a given time step $t$, the learning curve $L(t)$ gives an estimate of the average per-step reward of the model after $t$ timesteps were trained. It is calculated as an exponentially moving average with the previous rewards, calculated recursively as $L(t) = (1 - \theta)L(t-1) + \theta r_t$, where $r_t$ is the reward after the $t$-th step has been executed.[4] $\theta$ is a coefficient between 0 and 1, indicating the amount of smoothing on the pure rewards. We selected $\theta = 0.0003$.

Of course, for a recursive definition, we need to give a base case $L(0)$. For a specific problem, we introduce a default base case, the reward that an untrained agent usually yields. We use the same base case for all algorithms we compare on a specific environment. We decided to not just take the first reward as the base case, because the rewards have a high variance and our low $\theta$ puts heavy emphasis on the base case, especially in the early phasis of the training. For each configuration, we train 5 agents independently. In our plots, for a given time step, we show the interval between the minimum and maximum with a transparent color, and highlight the median with a line of the same color.

**Gym & Pendulum Environment**   The implementation was evaluated on several widespread standard-environments given by Gymnasium, the successor of OpenAI Gym (Brockman et al., 2016). Because the implementation is designed for continuous action spaces, we focus on standard and MuJoCo environments. First, we take a deep-dive with the `Pendulum-v1` environment. `Pendulum-v1` is an easy, but non-trivial environment with a three-dimensional observation space

---

[4]We do not use the cumulated episode reward because we are using a large number of environments, while only using a limited number of samples from each environment. Therefore, we do not have enough data to reliably estimate the episode lengths, especially in the early phasis, when many environments terminate early.

and a one-dimensional action space. The actions apply a torque on a pendulum to bring and keep it in an upright position. Negative reward is given depending on the angle that is missing towards the upright position and for higher absolute rotation speed. Ideally, the pendulum stays perfectly upright at zero speed during the whole episode. This would result in a perfect reward of zero. However, due to random initialization, the agent first needs to swing the pendulum into this position. This process is a nice benchmark for the trade-off between fast and long-term reward.

## 5.1 Direct Comparison between Factor Graph BNNs and PyTorch Neural Networks

First, we want to benchmark our algorithm against itself with standard NNs. As explained in Appendix C.1, BA2C can be operated with standard NNs. The whole logic is identical to the case with BNNs, including the calculation of the pseudo targets. We implemented an actor and critic via PyTorch (Paszke et al., 2019), which we use in our algorithm instead of the factor graph framework. The chosen architecture is the same like in Stable Baselines 3 (SB3), with two hidden layers of 64 neurons each, and tanh-activation function in between the layers.[5] As standard NNs do not propagate a standard deviation, the Python implementation usually returns $\beta_{actor}$ and $\beta_{critic}$ as constants.

Standard NNs and FG-BNNs can be set up to four combinations: (i) standard for both actor and critic, (ii) standard actor and Bayesian critic, (iii) vice versa, and (iv) both Bayesian. These four combinations were evaluated in Figure 1(a). Clearly, the agent improves its performance the fastest, if the factor graph NNs are used for both actor and critic (blue). If the actor gets a normal NN instead, the learning gets significantly slower, taking around the double amount of samples to reach a level of -2 (yellow). However, the training is still very stable, maxing out at practically the same level as the option with both NNs. If standard NNs are used for both actor and critic, the training gets significantly slower and less stable (green), eventually reaching the same performance on a longer training horizon (Figure 9 in Appendix E). Interestingly, the variant with a BNN as actor and a normal NN as critic does not improve to an acceptable level (green).

The comparison demonstrates that the BNN is more important for the critic than the actor, a phenomena worth investigating. We can explain the observation by plotting the mean squared error (MSE) of the critic network during training. As shown in Appendix E, the MSE is orders of magnitudes higher in both configurations with a normal NN as critic. The performance of the training is bottlenecked by the speed of the critic. That underlines the fact that the factor graph NNs learn faster, resulting in a lower error given the current policy. If the actor is also a normal NN, actor and critic learn at the same speed, resulting in slower, but still stable learning. If the actor is a BNN, the actor virtually "runs away" from the critic. As a result, the critic's value estimation is not up-to-date with the actor's policy, resulting in instability. The opposite situation, where the critic is Bayesian and the actor is normal, is stable, but slower. Notably, variants with a normal actor have a better asymptotic performance. They outperform the Bayesian variant at a certain point, but take longer to achieve this performance.

## 5.2 Does the BNN Exploration Mechanism help?

The previous experiment shows that the use of FG-BNNs indeed improves the training speed on the Pendulum environment. It is an interesting question, whether this improvement in training speed arises from the uncertainty-driven exploration of the BNNs compared to the fixed standard deviation which was used for the normal NNs, or if other properties are responsible. To test that, we set up a new experiment in which we benchmark three configurations: First, we use our BA2C agent in standard configuration (both actor and critic with FG-BNNs and uncertainty-driven exploration). Second, we use a BA2C, but instead of using the standard deviation obtained from the networks, we fix the standard deviations to $\beta_{actor}$, and $\beta_{critic}$ for actor and critic. Third, we use the configuration with standard NNs and fixed standard deviation as in the previous experiment. As we can see in Figure 1(b), learning speed is clearly better if the uncertainties are used for exploration. Nevertheless, BNNs still outperform normal NNs when no uncertainty-driven exploration happens.

---

[5]To allow exact comparability to the Bayesian implementation, the last layer of the actor does not squeeze the output into $[-1, 1]$ via tanh, but uses clipping as well. Check Appendix C.2.

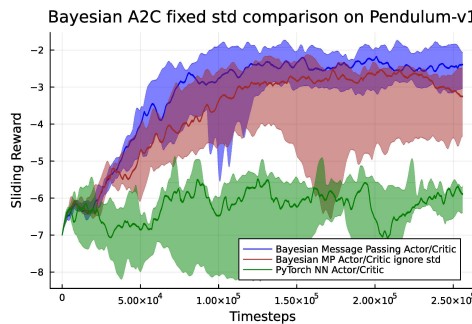

(a) This plot shows the learning curves of the RL agents on the Pendulum-v1 environment over 256,000 training steps. For both the actor and the critic, a standard NN and an FG-BNN can be used, resulting in four combinations; *Section 5.1*.

(b) Comparison of learning curves for Bayesian Actor Critic BA2C (blue), Bayesian Actor Critic BA2C with *fixed* standard deviation (red) and standard NNs (green); *Section 5.2*.

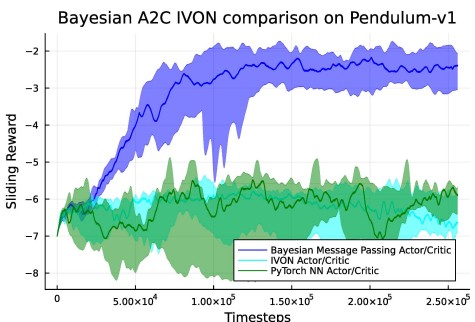

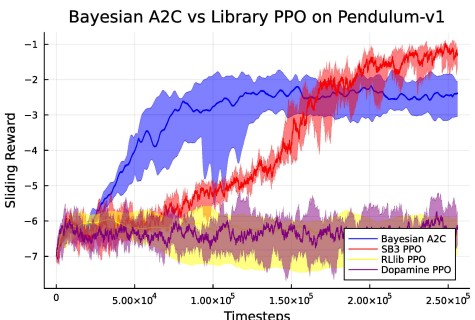

(c) The performance of our algorithm with IVON (turquise) for actor and critic against our standard configuration (blue) and standard NNs (green); *Section 5.3*.

(d) Comparison of our Bayesian actor critic BA2C with FG-BNNs (blue) against the PPO implementation of SB3 (red), Ray RLlib (yellow), and Google's Dopamine (purple); *Section 5.4*.

Figure 1: Performance Comparison of variants of BA2C, cf. (a)-(b), against IVON (c) and PPO (d) for Pendulum; for seven further Gymnasium environments with similar results, see Appendix F.

## 5.3 COMPARISON TO IVON

The choice of the factor graph framework to implement the agent is not straightforward, as there are several approaches for integrating uncertainty estimation into NNs. One widespread family of algorithms is variational inference (Zhang et al., 2018). Here, state of the art is IVON, showing significant advances even for large-scale NNs (Shen et al., 2024). Hence, IVON is a strong candidate for Bayesian RL as well, and it is interesting to test how the algorithm performs when trained with IVON for actor and critic. Here, we followed the hyperparameter guide of the IVON paper. With IVON for actor and critic, cf. Figure 1(c), the algorithm does not improve for Pendulum. In Appendix D, we did an ablation study over hyperparameters of IVON and found no configuration suitable. However, for other environments, we found that IVON indeed performed well (Figure 13).

## 5.4 LIBRARY COMPARISON

For a holistic comparison, the performance of our agents must be compared to state-of-the-art library implementations of policy-gradient actor-critic algorithms. For our comparison, we selected PPO implementations of three of the most popular RL libraries. SB3, cf. Raffin et al. (2021), is a widespread library and known for its high performance. The implementation is based on PyTorch. Because it is a direct successor of the original PPO implementation in the *Baselines*-library (Dhariwal et al., 2017), which has been unsupported since 2020, it is a good candidate for comparison. Ray RLlib (Liang et al., 2017) claims to be a "Industry-Grade, Scalable Reinforcement Learning" library, supporting distributed learning on large clusters. Ray RLlib supports both TensorFlow and PyTorch, but prefers PyTorch in their recent versions. Hence, we are using the PyTorch configuration.

Dopamine (Castro et al., 2018) is an RL framework open-sourced by Google and positions itself as "a research framework for fast prototyping of RL algorithms". It is based on TensorFlow. These implementations were three of the RL libraries with the most stars on GitHub (9.5k for SB3, 34.7k for the Ray framework, and 10.6k for Dopamine). So, they are a diverse and powerful group of competitors for our BA2C implementation. We ran all library algorithms with their given standard-configuration, but set $\gamma$ to 0.9 and the batch size to 256 to ensure same conditions for the library and our BA2C. On Pendulum, we verified that the selection of 0.9 instead of 0.99 for $\gamma$ improved the training speed of the standard PPO, and did not lower the final performance on the short horizon of early training. Figure 1(d) shows that especially SB3's PPO is a stable and reliable algorithm to solve Pendulum. It reaches the level of -3 after around 160,000 steps, which is around double the amount compared to the median of BA2C. agent, demonstrating a strong advantage of the Bayesian agent during early training. When looking at the level of -4, the advantage of BA2C is even stronger, where SB3 needs up to three times as many samples. However, in the later phasis of training, SB3 outperforms our Bayesian Agent in terms of peak-performance. Ray RLlib and Dopamine mostly fail to improve compared to the initial level.

Of course, our algorithm was tested on more environments than just Pendulum. The results are printed in the Appendix F. We will summarize the findings here. While SB3 often performs best in terms of final performance, the Bayesian algorithm and Ray RLlib are superior when it comes to early learning. The race between BA2C and Ray RLlib is close, with the Bayesian being faster on some environments, and Ray RLlib being faster on others. However, Ray RLlib is unable to solve Pendulum, and there is no environment where Ray RLlib clearly outperforms the Bayesian agent. Dopamine falls behind the other implementations.

## 6    LIMITATIONS, FUTURE WORK & CONCLUSION

**Limitations**    The actor critic algorithm with BNNs demonstrates stable and fast learning. However, the training speed was significantly reduced, while stability is also affected, and hyperparameter sensitivity is increased. Careful tuning, for example, with the GAE and reward and advantage normalization could help mitigating this problem. Nevertheless, a lower $\gamma$ does not hurt the value proposition of this algorithm, since it is designed for cases with sparse environment access, situations that often do not allow for a long lookahead.

The training with BNNs is more sample efficient, but is very time-consuming due to the high computation demand of the BNN framework. The training part of the algorithm takes around 50 times longer than with a PyTorch implementation. The longer computation time has two main reasons. First, the Bayesian frameworks, including the one we use, are results of recent research projects. In comparison to the highly optimized and professionally used PyTorch, they focus on a conceptual implementation instead of compute efficiency. Second, the probabilistic computations are just more complex mathematically, requiring more operations. Hence, a comparison with PyTorch in terms of resource efficiency is not expedient. Nevertheless, we think the optimization of Bayesian frameworks for better efficiency pose a worthy research opportunity.

**Future Work**    Although the Bayesian implementation provides significant improvements to the early training performance of on-policy algorithms, the sample efficiency of off-policy algorithms like SAC can be superior. In future research, one could utilize BNNs in the SAC algorithms to combine the benefits with regard to sample efficiency.

**Conclusion**    We proposed a variant of the A2C that works with BNNs which are not trained via gradients. Instead, we developed a pseudo-target formulation to train the actor. The use of BNNs improves the sample efficiency over standard NNs within the same algorithm significantly. When comparing the approach to highly mature RL implementations like SB3, the algorithm can prevail when it comes to the number of necessary samples to achieve competitive performance, and even outperforms the algorithm in the early phases.

The high training speed in the number of samples offers great opportunity for practitioners who need to work with sparse samples. For example, in robot control tasks, there is sometimes no accurate simulation environment available, and training is done with the real robot. In this case, the time required to optimize the parameters is minimal compared to the time for obtaining the samples. Here, our Bayesian approach can significantly improve training times.

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

# A   BACKGROUND: ADVANTAGE ACTOR CRITIC

A widespread RL algorithm is Advantage Actor Critic (A2C). This is an on-policy algorithm, which means that only samples obtained from the current policy can be used to train the agent. A2C uses two kinds of neural networks (NNs), an actor and a critic. Let us discuss the actor first, and motivate the critic later.

## A.1   ACTOR & POLICY GRADIENT THEOREM

For a given Markov decision process (MDP), let $\mathcal{S} = \mathbb{R}^n$ be an $n$-dimensional state space, and $\mathcal{A} = \mathbb{R}^m$ an $m$-dimensional action space. While A2C works both with discrete and continuous action spaces, we will entirely focus on continuous problems in this work. Actor Critic algorithms work with stochastic policies, so the policy $\pi$ of the RL agent gives a probability density function over the action space, given a current state:

$$\pi : \mathcal{S} \times \mathcal{A} \to [0, \infty) . \tag{14}$$

The actions applied in the environment in a given state are sampled from this distribution. In the vast majority of implementations for the continuous case, a Gaussian distribution is used for the policy function. Hence, the actor is an NN with weights $w$ that takes a state and outputs a mean and a standard deviation for each action dimension. In the multidimensional case, the actions are usually predicted independently (all covariances zero). We will write the means as $\mu_w(s)$ and the standard deviations as $\sigma_w(s)$.

The idea behind the training is intuitively easy to understand. During the training, actions are sampled, executed, and the reward is measured. If, in a specific state $s$ – a specific action $a$ turns out to be *good*, the probability is increased. If the action turns out to be *bad*, the probability is decreased. To increase or decrease the probability of $a$, the gradient of the probability of $a$ with respect to $w$ will be calculated, and gradient ascent is used to move the weights in the direction of the gradient.

The theoretical foundation for this is given by the policy gradient theorem from the literature. Let $J(w)$ denote the expected (discounted) return when following policy $\pi_w$ (for example, averaged over all relevant starting states). The policy gradient theorem states that

$$\nabla_w J(w) = \mathbb{E}_{s \sim d^\pi, \, a \sim \pi_w(\cdot|s)} \Big[ A(a, s) \, \nabla_w \log\big(\pi_w(a \mid s)\big) \Big], \tag{15}$$

where $d^\pi$ is the distribution of states under policy $\pi_w$, and $A(a, s)$ is the advantage of action $a$ in state $s$. In practice, we approximate this expectation by a single transition (or a batch of transitions), and perform a gradient ascent update of the form

$$w \; \leftarrow \; w \; + \; \alpha \, A(a, s) \, \nabla_w \log\big(\mathcal{N}\big(a; \mu_w(s), \mathrm{diag}(\sigma_w(s)^2)\big)\big). \tag{16}$$

Here, $\mathcal{N}(a; \, \mu_w(s), \sigma_w(s)^2)$ is the Gaussian probability density for action $a$ under the current policy parameters $w$, and $\alpha$ is the learning rate. This formula makes it clear that the probability of actions with positive advantage is increased, whereas it is decreased for actions with negative advantage.

## A.2   REWARD CALCULATION AND CRITIC

It is closely related to the reward $r$ the step gave, but it is more than that. The advantage measures the difference between the value (expected return) of the next state $V(s')$ and the current state $s$ (discounted):

$$A(a, s) = \big(r + \gamma V(s')\big) - V(s). \tag{17}$$

How do we know the state value? Here comes the critic into play. The critic is a neural network that calculates the function

$$V : \mathcal{S} \to \mathbb{R} \tag{18}$$

that maps a state to an estimate of its state value, the expected discounted sum of rewards from this step onward. It gets trained via bootstrapping, which means that predictions from the network itself for the subsequent step will be used for the update. Mathematically, the target for the updated state value $V^*(s)$ is

$$V^*(s) = r + \gamma V(s') . \tag{19}$$

With these formulas, we have everything needed to understand the vanilla version of Advantage Actor Critic (A2C).

## A.3 Exploration and Training

Finally, we need to discuss how exploration happens in A2C. Like every RL algorithm, it needs to tackle the trade-off between exploration and exploitation. How much should the algorithm focus to learn more about what it has already discovered, and how much should it try something else? Actor Critic does this in a quite elegant way: The stochastic policy assigns high probability to its preferred actions (the mean of the Gaussian distribution), but it always gives some probability to completely other actions. How much exploration happens is determined by the standard deviation parameter, which is an output of the NN. It gets trained via the gradient, just like the mean. During training, four cases can happen that are important to understand on an intuitive level:

1. The taken action is close to the mean (distance less than one standard deviation), and the advantage is positive: In this case, the training direction is to reduce the standard deviation, since this increases the probability density within those standard deviations.

2. The taken action is outside the range between the two inflection points and the advantage is positive: The standard deviation will be increased since this raises the probability density for the taken action.

3. If the action is inside the two inflection points and the advantage is negative, the standard deviation will be increased.

4. In the remaining case, the standard deviation will be decreased.

This change of the standard deviation is by no means arbitrary. It fully aligns with the policy gradient theorem, but it can be read as an auxiliary concept next to the desired training of the mean. How much exploration happens in a specific step does not have a real semantics, and regularly causes problems in practical applications. To tackle these problems, regularization is needed, for example an additional entropy loss in SAC.

## B Background: Approximate Message Passing for BNNs

The core objective in BNNs is to determine the predictive posterior distribution $p(y \mid x, D)$, representing the probability of an output $y$ given an input $x$ and a training dataset $\mathcal{D} = \{(x_i, y_i)\}_{i=1}^{n}$ of independent and identically distributed samples $(x_i, y_i)$. This distribution can be theoretically obtained by integrating over the network's weights $\theta$, which are treated as random variables with prior beliefs $p(\theta)$:

$$p(y \mid x, \mathcal{D}) = \int p(y \mid x, \theta) p(\theta \mid \mathcal{D}) \, d\boldsymbol{\theta}. \tag{20}$$

The posterior distribution of the weights, $p(\theta \mid \mathcal{D})$, is proportional to the product of the prior and the likelihood of $\mathcal{D}$, i.e.,

$$p(\theta \mid \mathcal{D}) \propto p(\theta) \prod_{i=1}^{n} p(y_i \mid f_\theta(x_i)), \tag{21}$$

where $f_\theta$ is the function realized by the network with the parameters $\theta$. The challenge lies in the intractability of the integral for complex networks.

Sommerfeld et al. (2025) address this challenge by representing the BNN as a factor graph. Factor graphs are probabilistic graphical models that facilitate the approximation of marginal distributions. The predictive posterior, as an integral over a product of factors, is well suited for this representation. However, direct modeling of the neural network function with Dirac delta functions leads to intractable message computations within the factor graph. Therefore, Sommerfeld et al. (2025) adopt a scalar-level representation, introducing latent variables and elementary Dirac delta factors to model the neural network's operations at a finer granularity. This decomposition allows for the derivation of MP equations, which are central to the belief propagation algorithm used to approximate marginals.

Due to the complexity of exact message calculations in large networks and datasets, and the presence of cycles in the factor graph, Sommerfeld et al. (2025) employ several key approximations. Firstly, messages are approximated as scaled Gaussian densities, leveraging the property that the product of Gaussians results in another Gaussian. This is facilitated by the use of the natural parameterization of Gaussians, where the precision and precision-mean are used instead of the mean and variance. Specifically, for a Gaussian $\mathcal{N}(\mu, \sigma^2)$, the precision is defined as $\rho = 1/\sigma^2$ and the precision-mean as

$\tau = \mu\rho$. This representation simplifies the multiplication of Gaussian densities and vastly improves the numerical stability of various message equations.

Secondly, moment matching is employed to approximate messages related to nonlinear activation functions. Direct moment matching is used where feasible. In cases where direct moment matching is difficult, a marginal approximation is used. This involves approximating the full marginal distribution of a variable with a Gaussian, and then using this approximation to derive an approximate message.

Thirdly, variational message passing is used to approximate the messages associated with the product of two variables. This approach, as described in Stern et al. (2009), helps in managing the symmetries present in the posterior distribution of BNNs.

The training procedure in Sommerfeld et al. (2025) uses loopy belief propagation with a specific message schedule to handle cyclic dependencies in the factor graph. A batching strategy is also employed to manage the computational cost of large datasets. This involves processing the training data in smaller batches and aggregating messages from inactive examples. Finally, prediction for unseen inputs is performed by propagating messages from the training branches to the prediction branch, effectively using the approximate posterior over weights as a prior during inference. This is an advantage over IVON which can only produce samples of the posterior predictive distribution via sampling from the posterior and doing a forward pass for each sample.

Lastly, the regression factor $\mathcal{N}(a; y, \beta^2)$ is of relevance for this paper. It represents the likelihood of observing a target value $y$ given an input $x$ and network parameters (or equivalently the networks output). Its role is crucial in updating the posterior distribution of the parameters during training. Formally, it connects two random variables $a$, a scalar outputted by the network, and $y$, the target. The variance $\beta^2$ is a constant (hyperparameter) we can interpret as aleatoric uncertainty or as a measure of belief we have in our target, i.e., epistemic uncertainty.

## C   DETAILS ON THE IMPLEMENTATION

### C.1   TRAINING IMPLEMENTATION

To test the impact of FG-BNNs, we developed an implementation of the concept above in Julia. The choice for Julia was made because the factor graph framework is implemented in Julia, and this programming language offers easy integration of common Python machine learning libraries. Our `Bayesian A2C Agent` is compatible with Gymnasium environments, the successor of OpenAI Gym. The current implementation only supports continuous action spaces.

In the main training loop, for each time step, the state of every environment is taken and used by the actor to predict the action mean and standard deviation. From these values, the action is sampled and executed, leading to new observations. Afterwards, the critic is run to estimate the values of the new and previous state, for a vanilla advantage and return estimation. While many production actor-critic implementations use the Generalized Advantage Estimator (GAE), we decided to go with the direct way for simplicity. Afterwards, the pseudo targets are calculated as described Section 3.2, and the parameter training happens for both actor and critic.

The implementation performs extensive logging of various metrics during training. After each batch, 24 measurements are written to Weights & Biases (Biewald, 2020). Examples are the outputs of the actor and critic, the differences between predicted action means and pseudo targets, the precisions and precision mean parameters from the inner workings of the FG-BNNs, and of course rewards. As most of these are empirical distributions among one batch (like the taken actions), or the parameters of the network, we save their mean, and 10, 25, 50, 75, and 90 percentiles. Rigorous logging also helps to explain the findings and differences.

The factor graph itself is not part of the `Bayesian A2C Agent` class, but accessed via a standard interface. For this interface, we needed to make slight changes to the factor graph library. The adapted code is part of the repository that is linked in this paper. The only communication between the agent and the factor graph happens through two function calls: *predict* returns the means and standard deviation for a batch, and *train_batch_new* is called with a state and a target (where we give the pseudo-target in case of the actor), together with the configuration parameters. This decoupling between the algorithm and the factor graph makes it possible to easily switch the type of the neural network to analyze the impact of the message-passing network compared to normal NNs and other

types of BNNs. In the experimental section, we will make use of this and run experiments with different kind of networks.

## C.2 Design Decisions for Training Stability

Just like for normal NNs, the samples in one batch should be independent and identically distributed in Bayesian networks. To satisfy this condition best, there are as many environments instantiated as the batch size says (in our case 256). Since many of our control problems have a fixed episode length, we perform a different number of pre-execution steps to each of the environments to make sure starts and ends are equally distributed among all batches. Additionally, in each step we reset an environment with a probability of $0.5\%$ to avoid the formation of similar environments in one batch. If the environments were synchronized in terms of timing, the samples would not be stochastically independent.

Research in the field of actor critic algorithms has shown that a common problem is overshooting while training the actor. When the parameters are optimized according to the policy gradient theorem, the policy often changes "too much", leading to unstable training. In response to that, algorithms like PPO aim to restrict the change of the policy. PPO clips the gradients to restrict the change of the action probability, slowing down the learning deliberately to improve stability. We adapt this idea by clipping the pseudo targets depending on the standard deviation given by the BNN. The hyperparameter $\kappa$ defines this percentage, defaulting to $1.25$.

In our analysis, we found that training stability could be improved by lowering the learning rate over the course of the training. Reducing the learning rate is a well-known technique in Machine Learning Smith (2018). Our algorithm uses an exponential decline, where the learning rate decreases exponentially with the number of steps, leveling off at $1/10$ of the initial learning rate.

Apart from these described design decisions, the implementation does not use tunings to keep it simple and put the focus on the change in function approximators.

## C.3 Notable Differences to Normal Neural Networks

While many RL algorithms use the tanh-function to squeeze the output into the interval of $[-1, 1]$, we do not use tanh and go for clipping instead. This decision was made because a tanh activation function would squeeze the normal distribution heavily that comes as a result of the last layer. Especially if the values are far away from zero, tanh would squeeze the variance close to zero, resulting in a stop of exploration and change. Clipping to the allowed action space happens both during inference and during calculation of the pseudo targets, ensuring that the pseudo targets are not outside of the action space.

The architecture of the BNN is close to the architecture proposed in the PPO paper, with two hidden layers of 64 neurons each. The first layer's input dimension is variable depending on the number of features of the selected environment, while the output of the critic has always one dimension, and the output of the actor has the same dimensionality as the action space. However, in contrast to the original implementation, the BNN uses LeakyReLU with a leak of $0.1$ as its activation function. LeakyReLU was found to perform better in for FG-BNNs, as discussed by Sommerfeld et al. (2025).

## C.4 Hyperparameters for Factor Graph Bayesian Neural Networks (FG-BNNs)

Furthermore, both the actor and the critic have a regression factor at their output position. This factor adds a default uncertainty to the prediction, resulting in two important hyperparameters $\beta_{actor}$ and $\beta_{critic}$. This hyperparameter sets a lower bound for the variance, which has the following two important consequences for the behavior of the algorithm:

- When it comes to the actor, this is a minimum level of exploration, even if the rest of the algorithm has converged. It is important to not select the value to low to ensure that training does not get stuck to early.
- It controls the training speed of the algorithm. A low variance on the output level means a high prior towards the predicted value during the backward pass when training the parameters. If the prior is too prejudiced, adjustment of the parameters will be considerably delayed.

Especially in the case of the critic, this is a problem if a critic that is too self-confident and, therefore, slow to learn cannot keep pace with changes to the policy.

Apart from $\beta_{actor}$ and $\beta_{critic}$, Actor Critic with BNNs requires some other choices of of hyperparameters than standard deep learning. The choices made for our benchmarks are shown in Table 1. For PPO, we used the standard parameters given in Table 2.

| Hyperparameter | Value | Description |
|---|---|---|
| *batch size* | 256 | The batch size for training equals the number of environments |
| *num batches* | 1 000 | It describes how many batches are trained before the training stops |
| $\gamma$ | 0.9 | The trade-off parameter between immediate and long-term reward |
| $\alpha$ | 0.05 | The "learning rate" in the Bayesian framework, effectively a scaling parameter for the advantage during pseudo-target calculation |
| *number factor graph iterations* | 3 | It describes how often forward & backward messages are sent during the training of one batch |
| $p_{reset}$ | 0.005 | A probability for resetting an environment at random |
| *action std expansion factor* | 4.0 | A regularization factor to foster exploration: the standard deviation of the actor gets multiplied by this before sampling an action during training |
| $\beta_{actor}$ | 0.05 | Minimum uncertainty of the actor prediction |
| $\beta_{critic}$ | 0.4 | Minimum uncertainty of the critic prediction |
| *action percentage clipping* | 0.9 | To improve stability, the pseudo targets are clipped to only 90% of the action space, ensuring that the side of the distribution also stays within the action space to a large percentage |
| $\kappa$ | 1.25 | Clip factor to limit change of the actor in an update step |
| $\alpha_{PyTorch}$ | 0.003 | The learning rate used in the PyTorch comparative implementation for both actor and critic |

Table 1: Hyperparameters of the Bayesian Actor Critic (BAC) algorithm with their default values. In the bottom section, the hyperparameters for the PyTorch comparative algorithm are given.

| Parameter | Value |
|---|---|
| Learning rate | $3 \cdot 10^{-4}$ |
| Steps per update | 2 048 |
| Minibatch size | 64 |
| Epochs per update | 10 |
| *clip_range* $(\varepsilon)$ | 0.2 |
| Discount factor $(\gamma)$ | 0.9 |
| Generalized advantage estimator factor $(\lambda)$ | 0.95 |
| Entropy coefficient | 0 |
| Value function coefficient | 0.5 |

Table 2: Hyperparameters for Proximal Policy Optimization (PPO).

## D  ABLATION STUDY

The hyperparameters given in Table 1 needed to be chosen with care in order to achieve good performance. We give a few examples for the Pendulum environment to justify that our selection is reasonable. However, due to the large space of possible hyperparameters, we were unable to do a fully comprehensive analysis. It may be well possible that a different choice of hyperparameters will give better results. That is especially true for the variety of environments. We did not do environment-specific hyperparameter tuning to get as general results as possible. Hence, the environments may benefit from additional tuning.

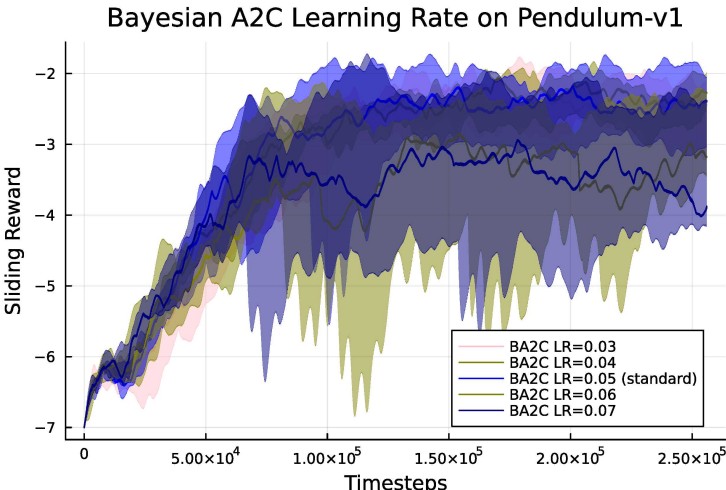

Figure 2: Learning curves for the BA2C algorithm with different learning rates on the Pendulum environment.

Figure 2 shows the learning curves of the BA2C algorithm on the Pendulum environment with different learning rates. The range of analyzed parameters is from 0.03 to 0.07, where 0.05 is our standard parameter. While these learning curves do not show completely different paths, there are noticeably differences. A lower learning rate tends to lead to slower learning, while the highest learning rate showed the highest initial training speed. However, the learning curves with a learning rate of 0.07 already tend to be more instable. This justifies our selection of 0.05 as our standard selection.

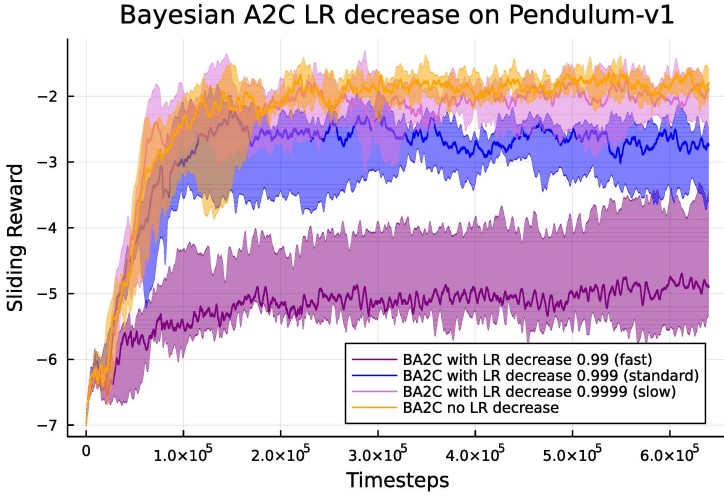

Figure 3: Learning curves of BA2C with different levels of decreasing learning rate on 640,000 steps.

Many machine learning implementations use a decreasing learning rate, so does our BA2C. We chose the exponential decline of the learning rate due to Figure 3. As you can see, the algorithm gets problems with stability if the learning rate remains unchanged, especially during later phases of training. Our standard parameter of 0.999 performs great in these experiments. If the learning rate is decreased faster, the initial learning rate will be to small, and no acceptable performance can be achieved.

When comparing the learning rate of the BA2C algorithm to the learning-rate hyperparameter in of neural networks trained with a classical gradient-based optimizer, please keep in mind that it has a very different semantic in BA2C. It is not a scaling factor for the gradients to directly influence the parameters, but it is the distance of the pseudo targets to the current prediction.

The following analysis of learning rates in the context of Adam or IVON neural networks refers to the learning rate given to the optimizer. The scaling factor on the distance of the pseudo targets stays at 0.05.

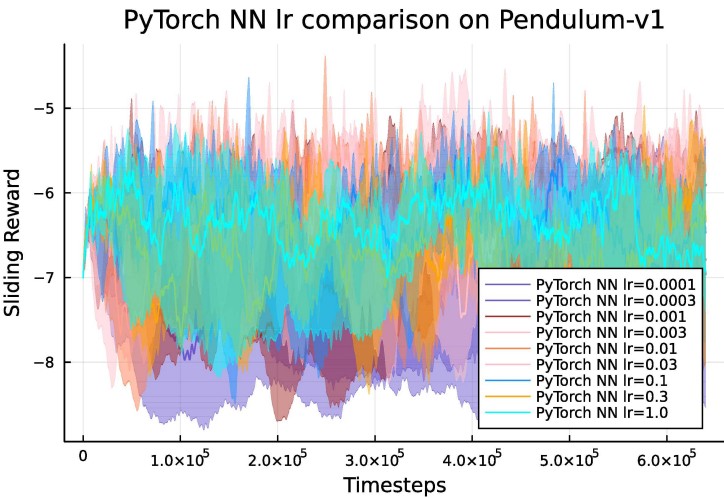

Figure 4: Comparison of different learning rates for PyTorch neural networks, selected for both actor and critic in our BA2C algorithm on 640,000 steps.

Next, we need to justify the learning rate of 0.003 for the PyTorch neural networks that are trained via Adam when they are used for actor and critic in our BA2C algorithm. First, 0.003 is also the chosen learning rate in Stable Baselines 3 for their neural networks of the same architecture, so we use an appropriate comparison here. Second, we checked the performance of the algorithm on the Pendulum environment while testing learning rates form 0.0001 to 1. The learning curves are visualized in Figure 4. As we can see, the chosen 0.003 clearly gives the best performance. To illustrate that it is actually able to reach a competitive performance, we show it for 640, 000 steps.

According to the hyperparameter guide in the appendix of the IVON paper Shen et al. (2024), we needed to set the effective sample size. In our case, this is the number of training steps. Additionally, we needed to set the learning rate parameter. We chose 0.1, a common selection for small neural networks. As you can see in Figure 13, IVON was relatively effective for some of the environments. On Pendulum however, IVON did not work at all. To check if the selection of the learning rate was responsible for this, we ran IVON with the same variety of learning rates like in 4. Unfortunately, Figure 13 shows that no learning rate worked for IVON on Pendulum.

We decided to implement the algorithm with advantage normalization. The primary reason to do this is to avoid the need of environment-specific learning rates. We analyzed the impact of this normalization on Pendulum and found no significant difference, when the rest of the configuration stays the same. Figure 6 shows the learning curves.

The next hyperparameter to justify is $\beta_{actor}$, the default uncertainty the algorithm carries. The default choice is 0.05. We tested a few parameters (shown in Figure 7) higher and lower, and found that the standard selection works best. The results for $\beta_{critic}$ are similar.

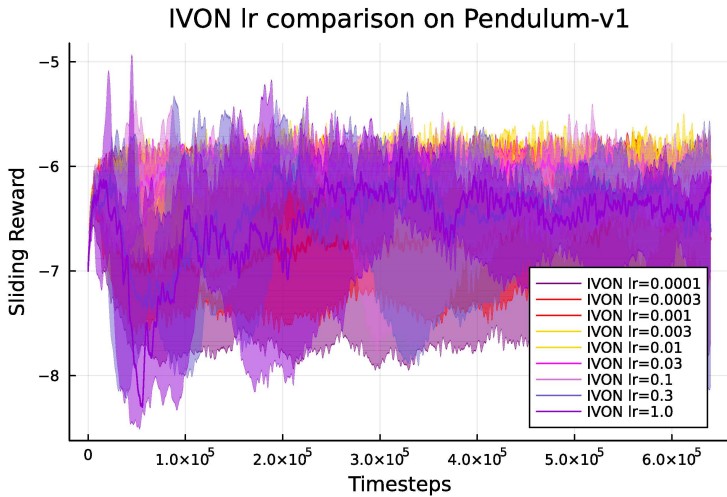

Figure 5: Comparison of different learning rates for BA2C training with IVON on 640,000 steps.

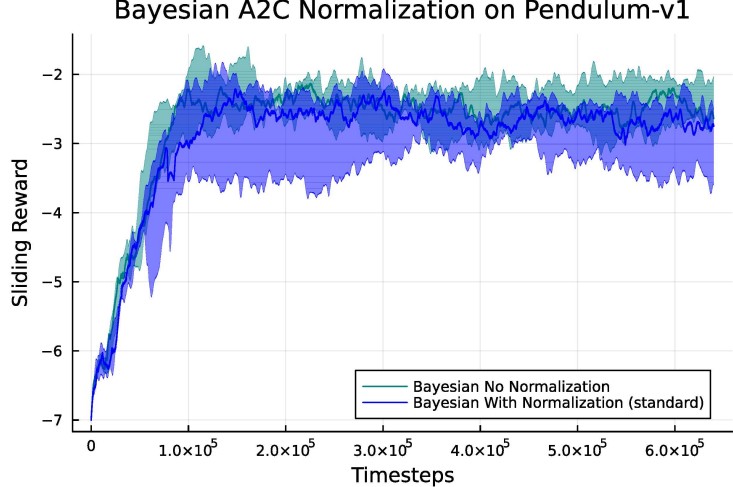

Figure 6: Learning curves of BA2C with and without advantage normalization.

To foster exploration, we use a linear scale between the uncertainty of the actor and the standard deviation of the stochastic, Gaussian policy. The standard choice of $4.0$ is not backed by theoretic findings, but by empiric results. Figure 8 shows a few different selections of this parameter around $4.0$.

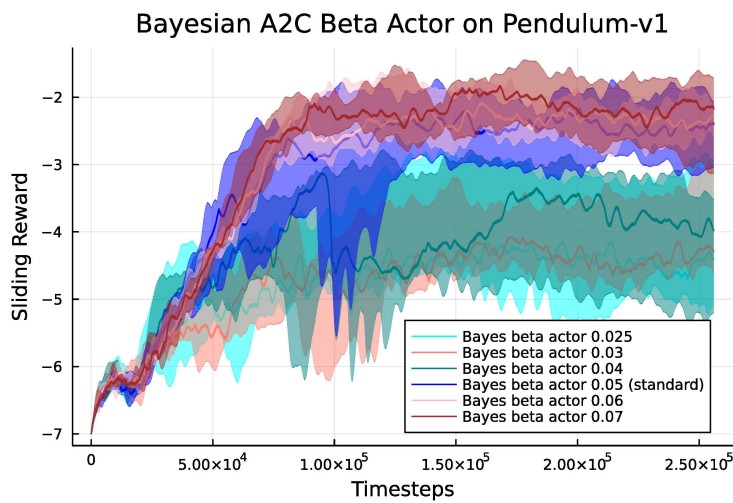

Figure 7: Different choices of $\beta_{actor}$ result in these learning curves.

Figure 8: Learning curves of the BA2C algorithm with different scaling factors from the uncertainty to the standard deviation of the Gaussian policy.

# E    ON THE FASTER LEARNING OF BA2C ON PENDULUM

Figure 9 shows the four combinations of FG-BNNs and PyTorch networks like Figure 1(a) in the main part. However, here we plot the curves over a longer time (640,000 samples) that illustrate the ability of the combination with two PyTorch networks to achieve competitive performance when more steps are granted.

For the same runs, Figure 10 shows the Mean Squared Error between the predicted state value of the critic and the target of the critic update. As we can see, this mean squared error is smaller by an order of magnitude for Bayesian critics (blue and yellow lines) compared to PyTorch critics (green and pastel-colored lines).

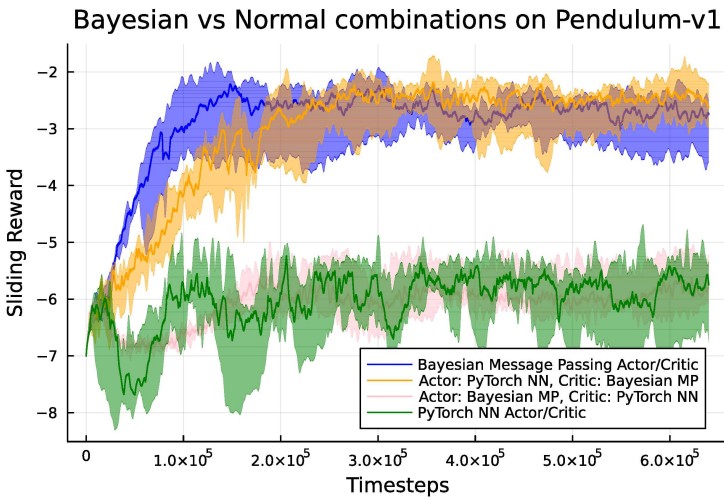

Figure 9: Learning Curves of BA2C combinations on Pendulum on a horizon of 640,000 steps.

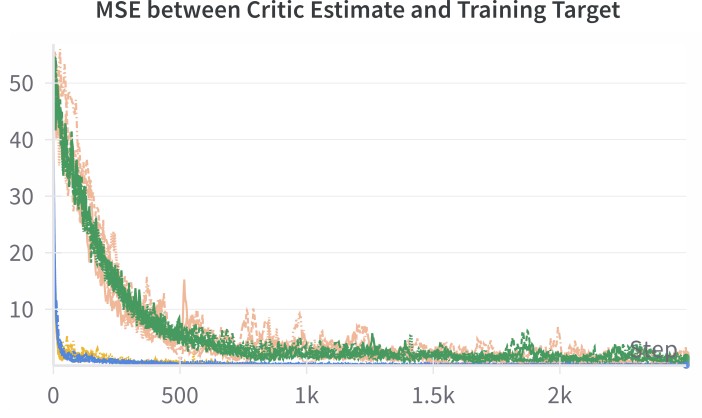

Figure 10: MSE between predicted state value by the critic and target: In the combinations with a Bayesian critic, the MSE decreases significantly faster, which is analogous to the finding in Figure 9, where agents with Bayesian critic achieved higher performance much quicker. Colors are also chosen like in Figure 9.

## F    ADDITIONAL EVALUATION ON OTHER GYMNASIUM ENVIRONMENTS

We continue the evaluation from Section 5 here. In the main paper, we analyzed the Pendulum environment with respect to different aspects. We compared FG-BNNs against PyTorch neural networks within the BA2C algorithm (Section 5.1), verified the importance of uncertainty-driven exploration (Section 5.2), tested the FG-BNNs against IVON (Section 5.3), and finally tested BA2C against popular PPO implementations. Here, we will add plots for the same experiments on 8 total environments (Pendulum plus seven other Gymnasium environments). Figure 11 shows how the four combinations of FG-BNNs and PyTorch neural networks perform on the eight environments. Figure 12 shows the comparison between networks with uncertainty-driven exploration and without. The comparison to IVON for the eight environments is given in Figure 13. In Figure 14, we provide the comparison of BA2C against the popular libraries.

While the plots are organized by experiment, the following text will walk through the results by environment. We will highlight and explain notable insights.

(a)  The Pendulum environment has been extensively studied in the main part.

(b) The Mountain Car environment is special because it is a setup with a hidden reward. The agent has to swing a car back and forth several times to climb a mountain. The reward is only given when the mountain has been successfully climbed, and each acceleration reduces the reward. An optimal agent would swing the car up the hill and accept the cost. However, Mountain Car is an environment that usually requires memorization (storing old experience). This would help the agent to learn from even very rare successes. Sometimes, we see upward-spikes in the learning curves for some of the agents. These upward-spikes arise if the environment was solved and a high reward was granted. However, all of our candidates are on-policy algorithms without memorization. Usually, they cannot learn from these rare successes, so they optimize to a policy with 0 actions. Our BA2C algorithms all find out about this very quickly. There are only marginal differences between the different networks and options within BA2C. When compared to the libraries, BA2C is the fastest algorithm to discovering the 0-policy, and achieves this policy practically instantaneous. Dopamine and SB3 take significantly longer. The reason for the lower final-performance of the Bayesian A2C is its forced level of higher exploration, causing costs for the taken actions.

(c) The Bipedal Walker environment is about moving a two-dimensional robot in a specific direction while controlling four joints. On this task, all libraries and our BA2C heavily struggle with stability, as it can be seen in Figure 14(c). This is not a surprise, given the 24-dimensional observation space and four-dimensional action space. An average reward of 0 means the agent can stabilize without falling, but does not move forward. This is a level that is only temporarily, never reliably achieved by all implementations. On the horizon of 50,000 steps, there is no clear winner. (On a much longer horizon, SB3 is able to improve significantly).

(d) The Inverted Pendulum environment can be considered the easiest out of these. BA2C achieves basically optimal performance, with few runs suffering from minor stability problems. Within the different options of implementing BA2C, we can confirm our findings from Pendulum: FG-BNNs work best, missing out on uncertainty-driven exploration hurts stability, and IVON gives acceptable but non-optimal performance. SB3 and Ray RLlib give high performance. SB3 is slightly behind in the beginning but achieves optimal performance most reliably, while Ray RLlib is fast in the beginning but not all runs get optimal. Dopamine also improves performance but does not get to a competitive level.

(e) We see a picture with very similar findings when testing Inverted Double Pendulum. One noticeable difference is in the comparison to the libraries (Figure 14(e)). Here, RLlib is a touch faster than BA2C in the very beginning, but BA2C outperforms RLlib after around $4,000$ samples. Stable Baselines 3 is significantly slower, but achieves significantly better performance after $12,500$ steps. One BA2C run became instable in a later phasis.

(f) On the Reacher environment, BA2C is clearly faster than all PPO libraries (Figure 14(f). However, it is interesting to see that the choice of FG-BNNs for both actor and critic is not optimal within the options for BA2C. As Figure 11(f) shows, the selection of PyTorch neural networks for the actor results in better performance, and so does IVON (Figure 11(f)). The reason might be the problem of FG-BNNs to handle multi-dimensional outputs well. Nevertheless, BA2C is a highly effective algorithm with one of the other options selected.

(g) The plot for Swimmer (Figure 14(g)) shows a remarkable hump at relatively early training for all variants of BA2C. We investigated the reason for this, and found that it is caused by the way the learning curves are designed, not by actual policy changes. The reason is that our BA2C runs with 256 environments in parallel. They are at a similar initialization-like state in the beginning. Then, the agent quickly finds out a movement that brings the swimmer forward by a small distance by contracting the arms. This is executed for all 256 environments. However, when the Swimmer is in the contracted state, it stays in this state, yielding zero rewards from now on. Unlike the other environments, Swimmer and Half Cheetah have very long episodes (a few thousand steps). The 256,000 steps that are shown in the learning curve however, are the result of round-robin scheduling of our environments, where only 1,000 steps are executed in each environment. Hence, BA2C's learning curve draws a pattern like a single evaluation on an environment. Why is this not a problem for the other environments? First, the episodes on the other environments are much shorter. Second, the pre-execution takes care of bringing environments with different progress into

one batch. But random sampling does not change the initial state of Swimmer much, while also running too few samples for bringing it close to its terminal state. Third, are less prone to entering deadlocks. For the stabilization tasks, the episode is often terminated early, and not stuck in an unhelpful state if it was entered. However, these aspects just impact the rendering of the plots, not the actual workings of the algorithms.

(h) On Half Cheetah, we see a similar hump like on Swimmer, and we see Half Cheetah being outperformed by the libraries Stable Baselines 3 and RLlib (Figure 14(h)). IVON and normal neural networks also do not work (Figure 13(h)), but we see remarkable performance of the combination of a Bayesian critic with a PyTorch actor in Figure 11(h) which is competitive to Stable Baselines 3. This indicates that there is indeed a huge potential if the problem of numeric instability of the factor-graph framework is solved. Moreover, this highlights the performance of our BA2C algorithm even on more complex environments.

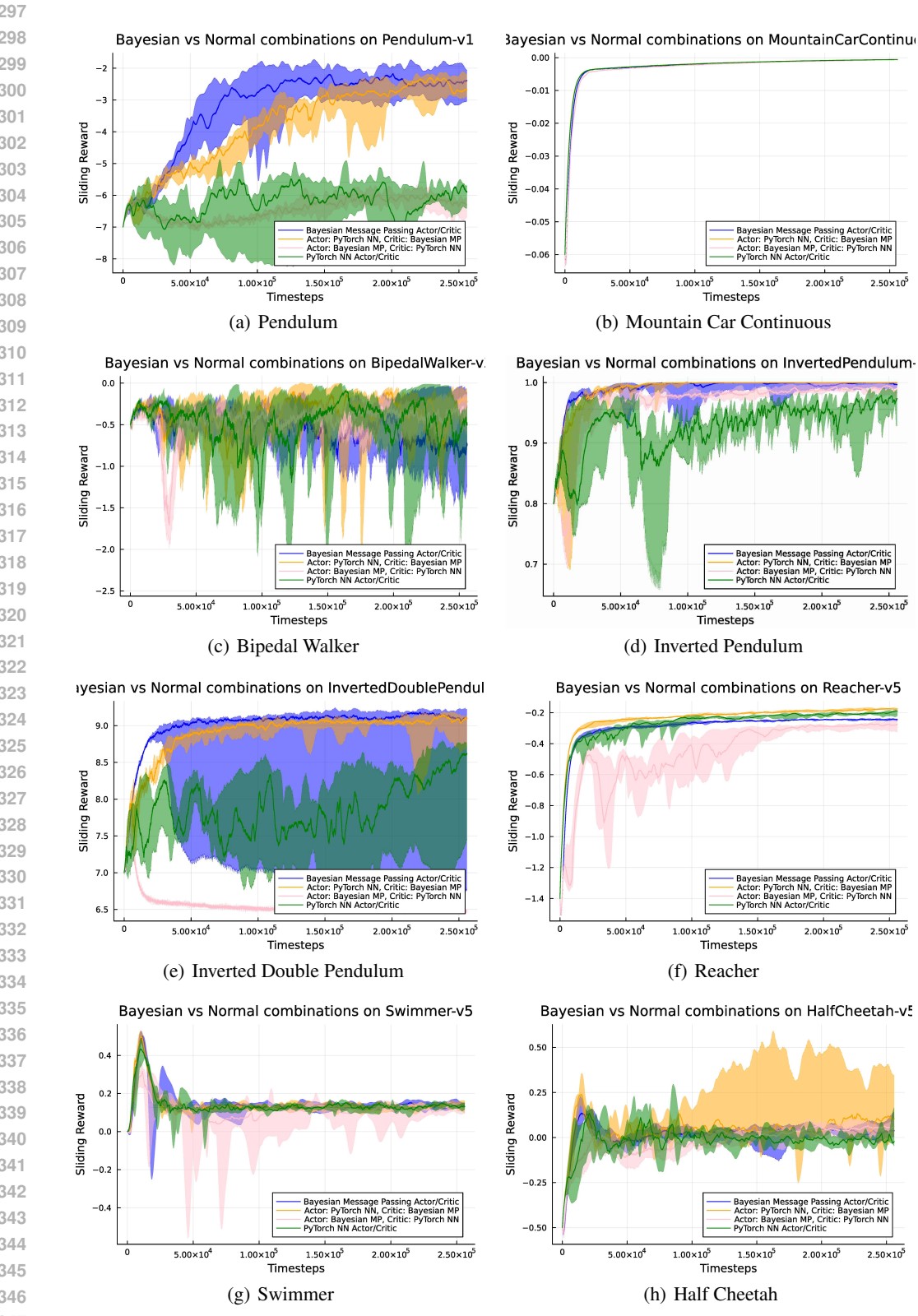

Figure 11: Combinations of Factor Graph & PyTorch.

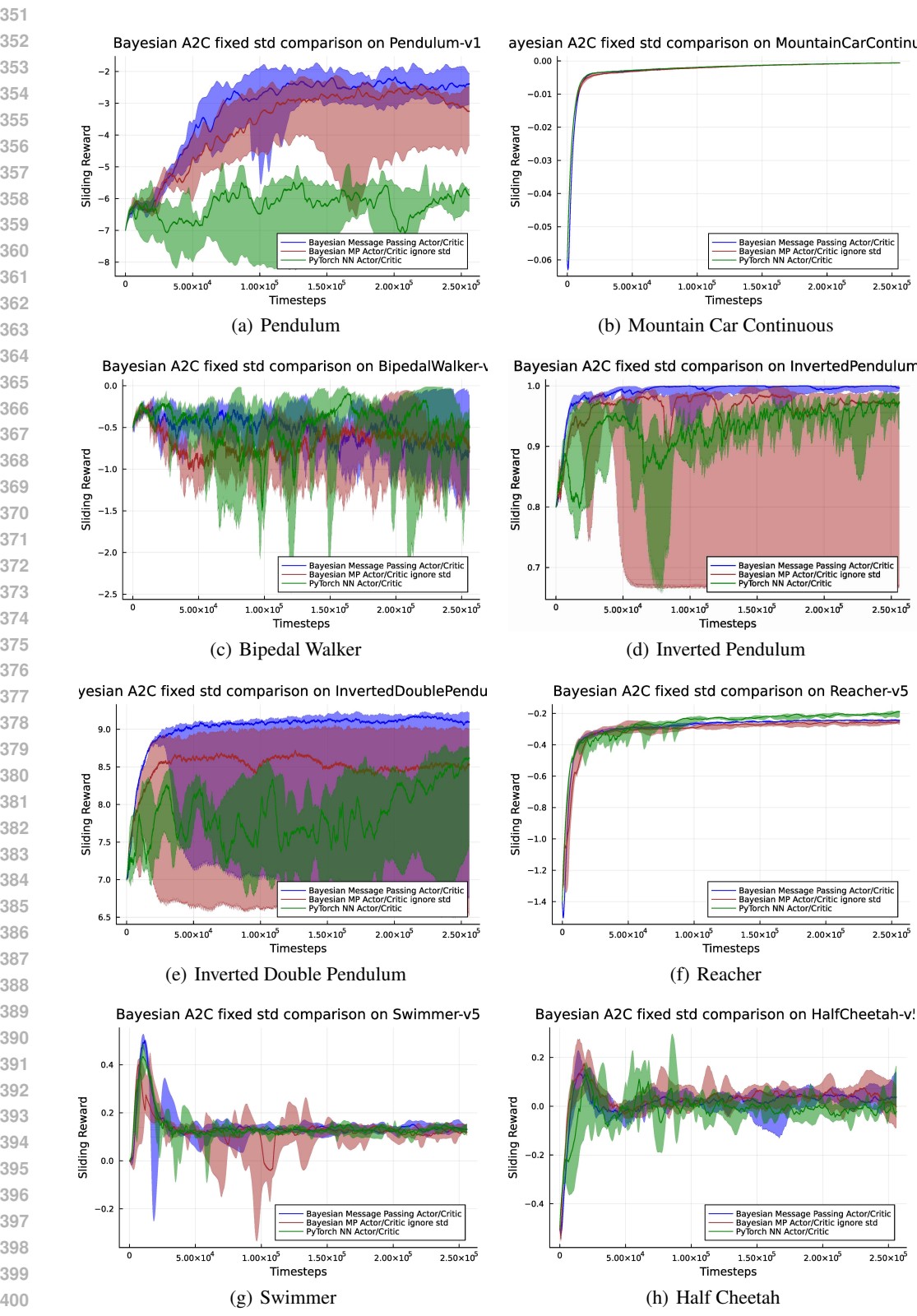

Figure 12: Factor Graph with Uncertainty-based Exploration vs. Fixed Exploration.

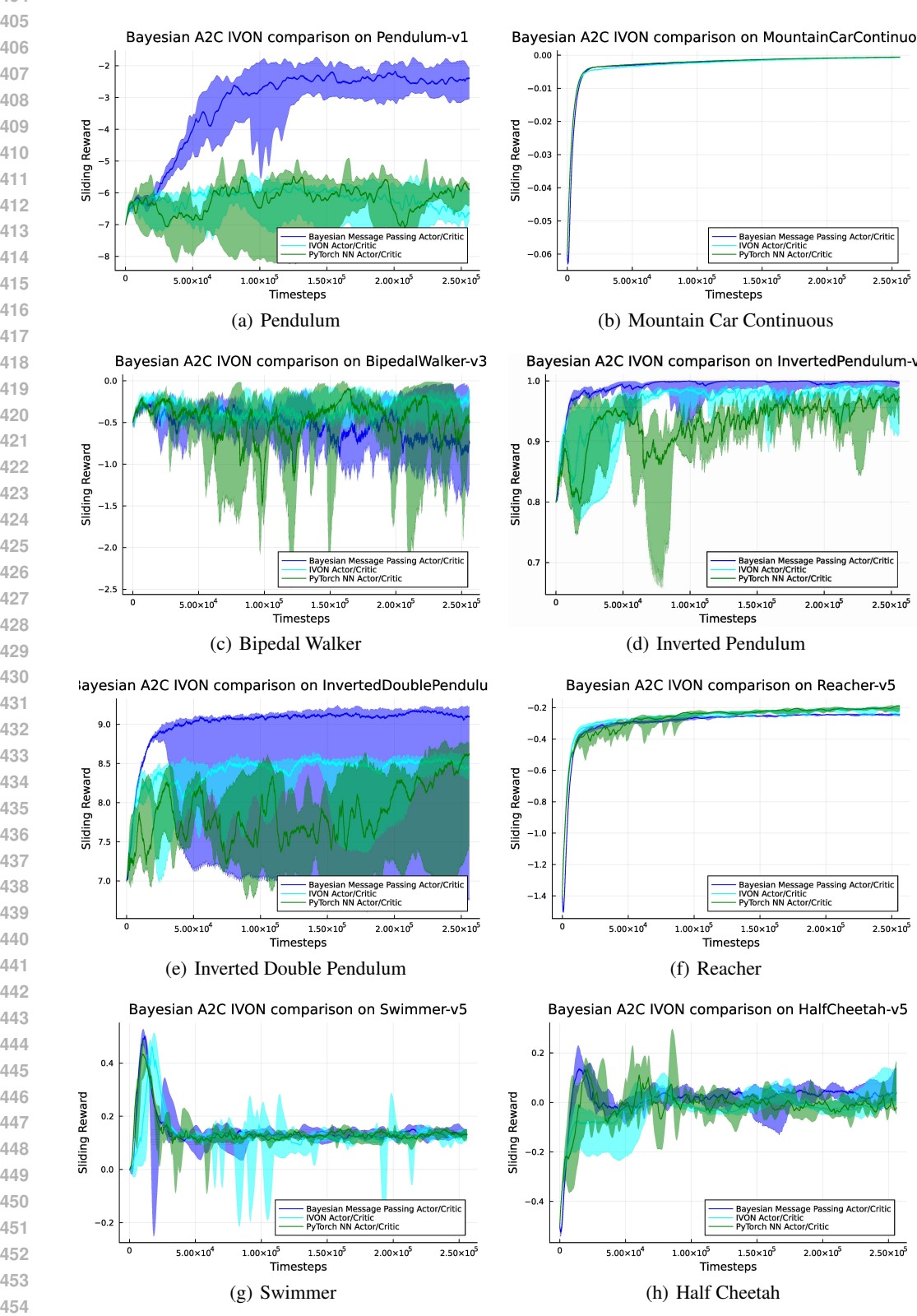

Figure 13: Learning curves of the eight environments to compare IVON against our standard BA2C and PyTorch neural networks.

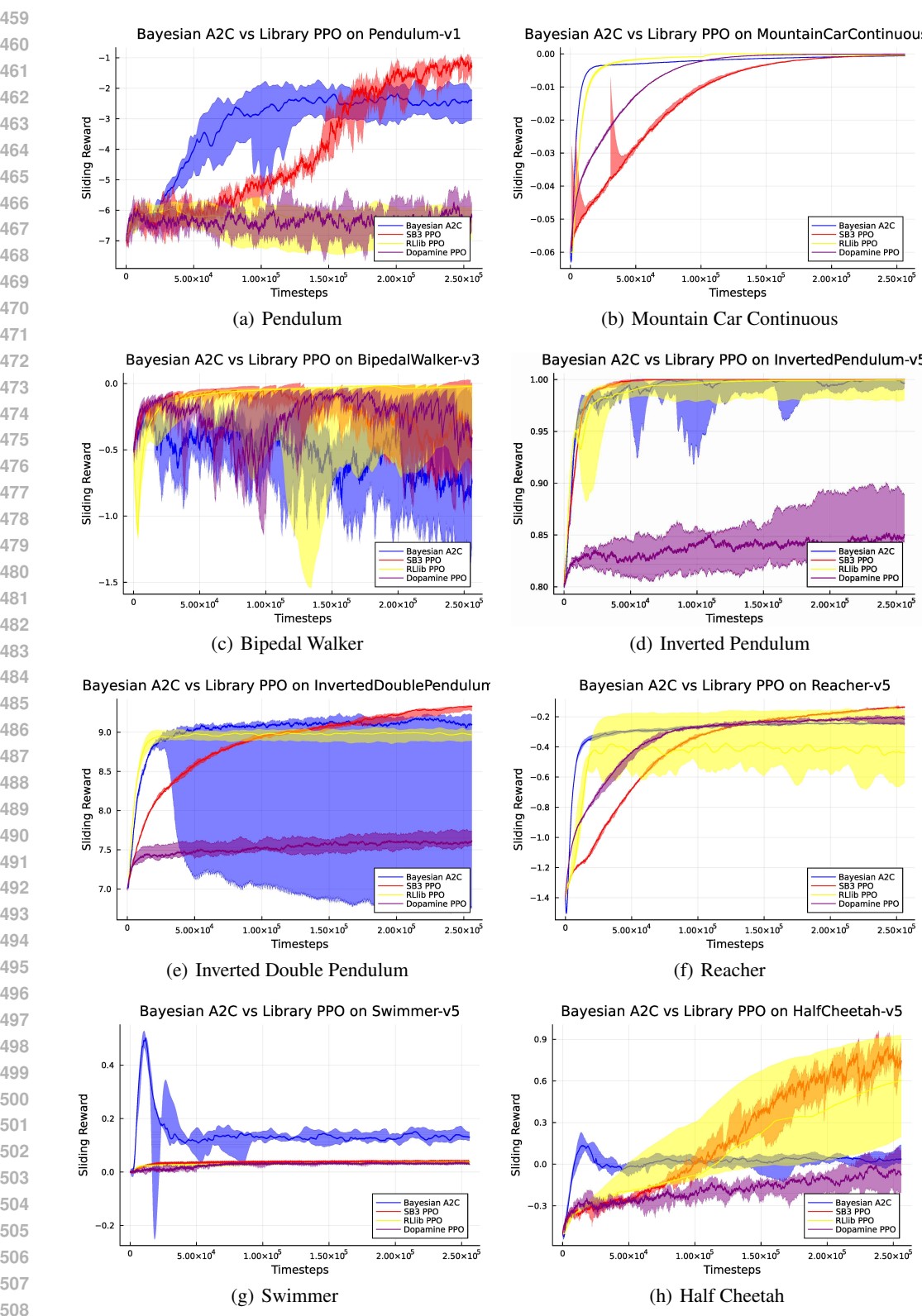

Figure 14: BA2C (blue) against PPO agents from three common libraries: SB3 (red), Ray RLlib (yellow), and Dopamine (purple).

