# OpenReview forum: "BA2C: Bayesian Advantage Actor Critic for Few Sample Learning using Factor Graph Bayesian Neural Networks"
_ICLR.cc/2026/Conference — Submitted to ICLR 2026_

### Official Review · Reviewer_Svb3 · 2025-10-26

**Soundness:** 2
**Presentation:** 2
**Contribution:** 2
**Rating:** 2
**Confidence:** 4

**Summary:**

Disclaimer: I have reviewed this paper previously. Since the paper didn't change since my last review I have not significantly changed it.

The paper proposes an on-policy RL algorithm using factor graph BNNs. Instead of policy gradients the actor is trained in a supervised fashion from sampled pseudo targets. The resulting algorithm is evaluated against standard PPO implementations. Further, the paper presents several ablation studies.

**Strengths:**

1. The use of Bayesian methods in RL is under-explored and this paper provides an interesting analysis of a recent inference method for BNNs in the context of RL.
1. The paper provides comparison against strong baseline implementations and several ablation studies.

**Weaknesses:**

1. The empirical evaluations are purely qualitatively with a big emphasis on the pendulum environment. The paper would greatly benefit from a systematic evaluation over many environments of the many interesting finding.

The evaluation of the method consists entirely of plots from learning curves. While these can give some qualitative insights in the learning behavior they alone are not a sufficient evaluation. In addition, the main paper only provides results for the pendulum environment. While this environment can provide a base case as a simple and low dimensional RL problem it alone is not a representative benchmark. I provide some suggestions for evaluation below.

2. A derivation of the pseudo targets, a main contribution, is missing.

The pseudo targets are defined, without derivation, in equation (11). While I can see the relation between equations (16) and (11) a formal derivation of the learning goal from the RL problem should be included in the paper.

3. While the paper presents several comparisons against on-policy methods it is missing comparisons against more sample-efficient baselines.

The paper only provides comparisons against PPO implementations and show improved sample-efficiency. However, PPO is not usually used for it's sample efficiency but instead for it's stability and low wall-clock times. Since BA2C has a significant higher wall-clock time and requires careful hyperparameter tuning I would suggest to also compare against stronger baselines such as off-policy algorithms and model-based RL.


**Suggestions to address Weakness 1:**

In RL we are, among other things, interested in sample efficiency and asymptotic performance. An evaluation of these metrics is missing from the paper. A possible way to evaluate sample-efficiency is to measure the (average) number of samples needed to reach a certain reward threshold (see [R4]). To estimate the asymptotic performance it is necessary to run the algorithms until approximate convergence. Learning curves that only show the initial training pahse are insufficient. I would recommend using a standard evaluation framework such as rliable [R3].

I would encourage the authors to include a pseudo code version of the proposed algorithm.

Since, in my opinion, the evaluation is the major weakness of the paper it is the main reason for my current score.

[R3] Agarwal, R., Schwarzer, M., Castro, P. S., Courville, A. C., & Bellemare, M. (2021). Deep reinforcement learning at the edge of the statistical precipice. Advances in neural information processing systems, 34, 29304-29320.

[R4] Mania, H., Guy, A., & Recht, B. (2018). Simple random search of static linear policies is competitive for reinforcement learning. Advances in neural information processing systems, 31.

**Questions:**

**Questions:**

1. The paper states that the method benefit from "uncertainty-driven exploration of the BNNs". In what sense is the exploration uncertainty driven? From what I understood the actions are sampled from the posterior distribution with no active exploration taking place.

2. The method assumes independent actions. This assumption is clearly violated in most environments. Why is this assumption necessary? Is it possible to relax this assumption to allow for covariance between actions?

3. I had some trouble following the training of the actor. I had a quick look at the supplementary code and saw that the actor is trained in supervised fashion on the mean of the pseudo target. Is that correct? Why is the variance of the actor fixed?

4. In your results it seems like the RLLib and Dopamine implementations of PPO are not able to learn anything on the pendulum environment. Why is that?

---

### Official Review · Reviewer_pHeT · 2025-10-30

**Soundness:** 2
**Presentation:** 2
**Contribution:** 2
**Rating:** 2
**Confidence:** 4

**Summary:**

This paper proposes a Bayesian advantage actor-critic algorithm, where the model is trained with expectation propagation for message passing, a gradient-free approach.

**Strengths:**

The strength of the work is that it poentially improves sample efficiency of RL algorithms.

**Weaknesses:**

1. I am puzzed that since the proposed algorithm is gradient-free, why is the Gassian assumption for the action, see equation (4), needed?

2. The description for Bayesian neural networks is vague.

3. How is the scalability of the proposed algorithm to big data problem?

4. The proposed method lacks theoretical guarantees for its performance.

5. The numerical experiments are limited.

**Questions:**

See also weakness:

1. I am puzzed that since the proposed algorithm is gradient-free, why is the Gassian assumption for the action, see equation (4), needed?

2. The description for Bayesian neural networks is vague.

3. How is the scalability of the proposed algorithm to big data problem?

4. The proposed method lacks theoretical guarantees for its performance.

---

### Official Review · Reviewer_ouPT · 2025-10-31

**Soundness:** 3
**Presentation:** 2
**Contribution:** 2
**Rating:** 4
**Confidence:** 3

**Summary:**

The paper presents a continuous actor-critic approach utilizing Bayesian factor graph networks. The uncertainty estimation ability of the Bayesian network allows for a natural implementation of sampling for exploration. The approach is evaluated and compared to baseline PPO algorithms on benchmark environments.

I am a bit split on this paper. The metholdogy is fair, and the contrbiution is straightforward (in a good way, as it is not unnecessary entangled, but also limiting novelty a bit). However, i feel that there are some weaknesses in the evaluation that need to be adressed.

**Strengths:**

+ The paper is well-written and comprehensible, and seems to be crafted with great care
+ The research problem is relevant, and the paper provides a fresh perspective beyond the standard approaches
+ The formulation of pseudo targets based on the RL objectives is a clear and straightforward contribution
+ I liked some of the experimental insights (e.g., the actor "running away from the Bayesian critic", section 5.1). The evaluation of different combinations of standard and Bayesian models, as well as the attributions, is an interesting series of experiments.

**Weaknesses:**

- The paper fails to engage with existing literature about uncertainty-guided exploration (and uncertainty estimation in general), e.g.
[Osband et al.] Deep Exploration via Bootstrapped DQN, 2016
[Zang et al.] Proximal Policy Optimization via Enhanced Exploration Efficiency, 2020
[Charpentier et al.] Disentangling epistemic and aleatoric uncertainty in reinforcement learning, 2022
- In particular, I would like a comparison with some additional simple approximate Bayesian baselines, e.g., deep ensembles, Monte-Carlo Dropout, etc., for uncertainty estimation
- I would have liked a discussion of the most important design choices (section 4) within the main paper. Pushing everything to the Appendix leaves a lot of work to the reader (also due to the fact that there is a lot of text in the appendices), instead of clearly prioritizing
- The main experiments are just being conducted on a single environment, which significantly limits the expressiveness of the experiments. Furthermore, many hyperparameters like learning rate, network sizes, re-sampling of entries (n_epochs  in SB3 PPO), etc., are not ablated in the baseline algorithms (although LR in particular can have a huge impact on learning speed). Figure 11 in the Appendix shows more inconsistent results (although BA2C still seems to perform the best)

Minor:
- The description of the learning curve (5 Evaluation Methods) is very thorough in the main paper, although it is a very common plot. I would probably shorten it (and move it to the Appendix) and instead extend the implementation section. Also, it is pretty common to evaluate the policy in a separate environment (without gradient updates), so measuring the entire episode return would be very much possible by rolling out the policy during training
- The library comparison takes a lot of space in the main paper, but I did not find it as insightful (or the be presise a different kind of issue. The performance differences between libraries are interesting..but not the part of the main contribution of this paper). I would rather see it replaced with results for additional environment, ablation experiments, etc. (A comparison with just the SB3 PPO baseline seems appropriate)

**Questions:**

- Do you have an explanation for the subpar performance of Dopamine and RayLib PPO? Are there differences in implementation? Is this a known issue?

---

### Official Review · Reviewer_WDig · 2025-11-01

**Soundness:** 3
**Presentation:** 2
**Contribution:** 2
**Rating:** 6
**Confidence:** 2

**Summary:**

This paper proposes a novel Bayesian reinforcement learning algorithm called Bayesian Advantage Actor Critic (BA2C), which leverages factor graph-based Bayesian neural networks (FG-BNNs) trained via approximate message passing (AMP). The core idea is to improve sample efficiency in on-policy RL - particularly in early training stages - by using uncertainty estimates from BNNs to guide exploration.

**Strengths:**

1. **Novel and Well-Motivated Approach:** The integration of factor graph BNNs with AMP into an on-policy actor-critic framework is innovative. The use of uncertainty quantification for exploration is conceptually sound and well aligned with Bayesian RL principles.
2. **Strong Empirical Evaluation:** The authors implement BA2C in Julia, using expectation propagation for inference, and compare it to standard neural networks (via PyTorch), IVON (a state-of-the-art variational inference method), and popular PPO implementations (Stable Baselines 3, Ray RLlib, Dopamine). Experiments on Gymnasium environments, especially Pendulum-v1, demonstrate that BA2C outperforms neural and PPO baselines on Gymnasium tasks.
3. **Clear Focus on Sample Efficiency:** The paper directly addresses a critical challenge in real-world RL: limited data availability. The demonstrated 50% reduction in required samples for early convergence makes this work highly relevant for applications like robotics and control systems.

**Weaknesses:**

1. **Computational Overhead and Scalability:** While the paper demonstrates superior sample efficiency, it does not fully address the trade-off in computational cost. Training with FG-BNNs via AMP is significantly slower than PyTorch (reported to be ~50× slower). This could limit the practical applicability in time-sensitive real-world scenarios. The paper should discuss this trade-off more explicitly and consider future directions for efficiency improvements.
2. **Limited Discussion on Hyperparameter Sensitivity:** Although an ablation study is included, the paper could benefit from a deeper discussion of hyperparameter sensitivity. For example, how robust is the performance of BA2C across different environments with varying levels of noise or complexity? A more systematic sensitivity analysis would strengthen the claims.
3. **Lack of Statistical Significance Testing:** While the results are presented with median and confidence intervals, the paper does not perform formal statistical tests (e.g., t-tests or ANOVA) to assess whether differences between BA2C and baselines are statistically significant. Including such analysis would improve the rigor of the evaluation.
4. **Potential Overfitting to Pendulum Environment:** The most compelling results are shown on Pendulum-v1. While the paper validates results on 8 environments, the visualizations and discussion are most detailed for Pendulum. It would be beneficial to include a more balanced analysis of performance across all environments, especially those where BA2C underperforms (e.g., Half Cheetah).
5. **Clarification Needed on Pseudo-Target Derivation:** The derivation of the pseudo-target in Section 3.2 is mathematically sound but not fully intuitive. A more explanatory paragraph or a small diagram could help readers understand why this form of pseudo-target is a valid proxy for the policy gradient in a non-gradient-based framework.

**Questions:**

1. **Computational Efficiency:** Given that AMP-based training is ~50× slower than PyTorch, how do you envision BA2C being deployed in real-world applications where training time is a constraint? Could the proposed method be combined with techniques like distillation or distillation-enabled transfer learning to reduce inference cost?

2. **Generalization Across Environments:** The paper shows that BA2C performs exceptionally well on Pendulum and Inverted Pendulum, but struggles on tasks like Swimmer and Half Cheetah. What do you believe are the key differences between these environments that could explain the performance gap? Is this related to the dimensionality of the action space or the nature of the dynamics?

3. **Hyperparameter Tuning:** Could you provide insight into the tuning process for the hyperparameters (e.g., βactor, βcritic, learning rate α)? Were these tuned on a single environment (e.g., Pendulum), and then applied uniformly? How sensitive is the performance to these choices across different environments?

4. **Comparison to Other Bayesian Methods:** The paper compares to IVON, but there are other Bayesian deep RL methods (e.g., Bayesian Q-learning, Monte Carlo dropout-based exploration). Could you briefly discuss how BA2C compares to these alternatives in terms of sample efficiency and stability?

5. **Future Work on Stability:** The paper notes that numerical instability in factor graph frameworks limits performance. What are your plans for addressing this issue in future work? Are there promising directions such as improved message-passing schedules, better numerical conditioning, or hybrid gradient-AMP training?

---

### Meta-Review · Area_Chair_PR3q · 2025-12-26

**Summary:**

This paper proposes an A2C algorithm with Bayesian neural network training that supposedly improves exploration and learning efficiency by epxloiting epistemic uncertainty.
The experimental evalation is mostly limited to a pendulum envirionment with weak baselines, and the algorithm is not sufficiently motivated.
There is no rebuttal.

**Reviewer Concerns:**

I strongly agree with the review of Reviewer Svb3. The paper in its current form is way below the bar for ICLR. The general topic is of interest, and I encourage the authors to improve their work.

**Reviewer Scores:**

6,4,2,2
But the 6 looks very much LLM generated.

---

### Decision · Program_Chairs · 2026-01-26

Reject